# Tight Bounds for Collaborative PAC Learning via Multiplicative Weights[*]

**Jiecao Chen**
Computer Science Department
Indiana University at Bloomington
jiecchen@iu.edu

**Qin Zhang**
Computer Science Department
Indiana University at Bloomington
qzhangcs@indiana.edu

**Yuan Zhou**
Computer Science Department
Indiana University at Bloomington
and
Department of Industrial and Enterprise Systems Engineering
University of Illinois at Urbana-Champaign
yuanz@illinois.edu

## Abstract

We study the collaborative PAC learning problem recently proposed in Blum et al. [3], in which we have $k$ players and they want to learn a target function collaboratively, such that the learned function approximates the target function well on all players' distributions simultaneously. The quality of the collaborative learning algorithm is measured by the ratio between the sample complexity of the algorithm and that of the learning algorithm for a single distribution (called the overhead). We obtain a collaborative learning algorithm with overhead $O(\ln k)$, improving the one with overhead $O(\ln^2 k)$ in [3]. We also show that an $\Omega(\ln k)$ overhead is inevitable when $k$ is polynomial bounded by the VC dimension of the hypothesis class. Finally, our experimental study has demonstrated the superiority of our algorithm compared with the one in Blum et al. [3] on real-world datasets.

## 1  Introduction

In this paper we study the collaborative PAC learning problem recently proposed in Blum et al. [3]. In this problem we have an instance space $\mathcal{X}$, a label space $\mathcal{Y}$, and an unknown target function $f^*$ : $\mathcal{X} \to \mathcal{Y}$ chosen from the hypothesis class $\mathcal{F}$. We have $k$ players with distributions $D_1, D_2, \ldots, D_k$ labeled by the target function $f^*$. Our goal is to *probably approximately correct (PAC)* learn the target function $f^*$ for *every* distribution $D_i$. That is, for any given parameters $\epsilon, \delta > 0$, we need to return a function $f$ so that with probability $1 - \delta$, $f$ agrees with the target $f^*$ on instances of at least $1 - \epsilon$ probability mass in $D_i$ for every player $i$.

As a motivating example, consider a scenario of personalized medicine where a pharmaceutical company wants to obtain a prediction model for dose-response relationship of a certain drug based on the genomic profiles of individual patients. While existing machine learning methods are efficient to learn the model with good accuracy for the whole population, for fairness consideration, it is also desirable to ensure the model accuracies among demographic subgroups, e.g. defined by gender, ethnicity, age, social-economic status and etc., where each of them is associated with a label distribution.

---

[*]A full version of this paper is available at https://arxiv.org/abs/1805.09217

We will be interested in the ratio between the sample complexity required by the best collaborative learning algorithm and that of the learning algorithm for a single distribution, which is called the *overhead* ratio. A naïve approach for collaborative learning is to allocate a uniform sample budget for each player distribution, and learn the model using all collected samples. In this method, the players do minimal collaboration with each other and it leads to an $\Omega(k)$ overhead for many hypothesis classes (which is particularly true for the classes with fixed VC dimension – the ones we will focus on in this paper). In this paper we aim to develop a collaborative learning algorithm with the optimal overhead ratio.

**Our Results.** We will focus on the hypothesis class $\mathcal{F} = \{f : \mathcal{X} \to \mathcal{Y}\}$ with VC dimension $d$. For every $\epsilon, \delta > 0$, let $\mathbb{S}_{\epsilon,\delta}$ be the sample complexity needed to $(\epsilon, \delta)$-PAC learn the class $\mathcal{F}$. It is known that there exists an $(\epsilon, \delta)$-PAC learning algorithm $\mathcal{L}_{\epsilon,\delta,\mathcal{F}}$ with $\mathbb{S}_{\epsilon,\delta} = O\left(\frac{1}{\epsilon}\left(d + \ln \delta^{-1}\right)\right)$ [10]. We remark that we will use the algorithm $\mathcal{L}$ as a blackbox, and therefore our algorithms can be easily extended to other hypothesis classes given their single-distribution learning algorithms.

Given a function $g$ and a set of samples $T$, let $\mathrm{err}_T(g) = \mathbf{Pr}_{(x,y)\in T}[g(x) \neq y]$ be the *error* of $g$ on $T$. Given a distribution $D$ over $\mathcal{X} \times \mathcal{Y}$, define $\mathrm{err}_D(g) = \mathbf{Pr}_{(x,y)\sim D}[g(x) \neq y]$ to be the *error* of $g$ on $D$. The $(\epsilon, \delta)$-PAC $k$-player collaborative learning problem can be rephrased as follows: For player distributions $D_1, D_2, \ldots, D_k$ and a target function $f^* \in \mathcal{F}$, our goal is to learn a function $g : \mathcal{X} \to \mathcal{Y}$ so that $\mathbf{Pr}[\forall i = 1, 2, \ldots k, \mathrm{err}_{D_i}(f^*, g) \leq \epsilon] \geq 1 - \delta$. Here we allow the learning algorithm to be *improper*, that is, the learned function $g$ does not have to be a member of $\mathcal{F}$.

Blum et al. [3] showed an algorithm with sample complexity $O\left(\frac{\ln^2 k}{\epsilon}\left((d + k)\ln \epsilon^{-1} + k \ln \delta^{-1}\right)\right)$. When $k = O(d)$, this leads to an overhead ratio of $O(\ln^2 k)$ (assuming $\epsilon, \delta$ are constants). In this paper we propose an algorithm with sample complexity $O\left(\frac{(\ln k + \ln \delta^{-1})(d+k)}{\epsilon}\right)$ (Theorem 4), which gives an overhead ratio of $O(\ln k)$ when $k = O(d)$ and for constant $\delta$, matching the $\Omega(\ln k)$ lower bound proved in Blum et al. [3].

Similarly to the algorithm in Blum et al. [3], our algorithm runs in rounds and return the plurality of the functions computed in each round as the learned function $g$. In each round, the algorithm adaptively decides the number of samples to be taken from each player distribution, and calls $\mathcal{L}$ to learn a function. While the algorithm in Blum et al. [3] uses a grouping idea and evenly takes samples from the distribution in each group, our algorithm adopts the multiplicative weight method. In our algorithm, each player distribution is associated with a weight which helps to direct the algorithm to distribute the sample budget among all player distributions. After each round, the weight for a player distribution increases if the function learned in the round is not accurate on the distribution, letting the algorithm pay more attention to it in the future rounds. We will first present a direct application of the multiplicative weight method which leads to a slightly worse sample complexity bound (Theorem 3), and then prove Theorem 4 with more refined algorithmic ideas.

On the lower bound side, the lower bound result in Blum et al. [3] is only for the special case when $k = d$. We extend their result to every $k$ and $d$. In particular, we show that the sample complexity for collaborative learning has to be $\Omega(\max\{d \ln k, k \ln d\}/\epsilon)$ for constant $\delta$ (Theorem 6). Therefore, the sample complexity of our algorithm is optimal when $k = d^{O(1)}$. [2]

Finally, we have implemented our algorithms and compared with the one in Blum et al. [3] and the naïve method on several real-world datasets. Our experimental results demonstrate the superiority of our algorithm in terms of the sample complexity.

**Related Work.** As mentioned, collaborative PAC learning was first studied in Blum et al. [3]. Besides the problem of learning one hypothesis that is good for all players' distributions (called the *centralized collaborative learning* in [3]), the authors also studied the case in which we can use different hypotheses for different distributions (called *personalized collaborative learning*). For the personalized version they obtained an $O(\ln k)$ overhead in sample complexity. Our results show that

we can obtain the same overhead for the (more difficult) centralized version. In a concurrent work [15], the authors showed the similar results as in our paper.

Both our algorithms and Adaboost [7] use the multiplicative weights method. While Adaboost places weights on the samples in the prefixed training set, our algorithms place weights on the distributions of data points, and adaptively acquire new samples to achieve better accuracy. Another important feature of our improved algorithm is that it tolerates a few "failed rounds" in the multiplicative weights method, which requires more efforts in its analysis and is crucial to shaving the extra $\ln k$ factor when $k = \Theta(d)$.

Balcan et al. [1] studied the problem of finding a hypothesis that approximates the target function well on the joint mixture of $k$ distributions of $k$ players. They focused on minimizing the communication between the players, and allow players to exchange not only samples but also hypothesis and other information. Daume et al. [11, 12] studied the problem of computing linear separators in a similar distributed communication model. The communication complexity of distributed learning has also been studied for a number of other problems, including principal component analysis [13], clustering [2, 9], multi-task learning [16], etc.

Another related direction of research is the multi-source domain adaption problem [14], where we have $k$ distributions, and a hypothesis with error at most $\epsilon$ on each of the $k$ distributions. The task is to combine the $k$ hypotheses to a single one which has error at most $k\epsilon$ on any mixture of the $k$ distribution. This problem is different from our setting in that we want to learn the "global" hypothesis from scratch instead of combine the existing ones.

## 2 The Basic Algorithm

In this section we propose an algorithm for collaborative learning using the multiplicative weight method. The algorithm is described in Algorithm 1, using Algorithm 2 as a subroutine.

We briefly describe Algorithm 1 in words. We start by giving a unit weight to each of the $k$ player. The algorithm runs in $T = O(\ln k)$ rounds, and players' weights will change at each round. At round $t$, we take a set of samples $S^{(t)}$ from the average distribution of the $k$ players weighted by their weights. We then learn a classifier $g^{(t)}$ for samples in $S^{(t)}$, and test for each player $i$ whether $g^{(t)}$ agrees with the target function $f^*$ with probability mass at least $1 - \epsilon/6$ on distribution $D_i$. If yes then we keep the weight of the $i$-th player; otherwise we multiply its weight by a factor of 2, so that $D_i$ will attract more attention in the future learning process. Finally, we return a classifier $g$ which takes the plurality vote[3] of the $T$ classifiers $g^{(0)}, g^{(1)}, \ldots, g^{(T-1)}$ that we have constructed. We note that we make no effort to optimize the constants in the algorithms and their theoretical analysis; while in the experiment section, we will tune the constants for better empirical performance.

The following lemma shows that TEST returns, with high probability, the desired set of players where $g$ is an accurate hypothesis for its own distribution. We say a call to TEST *successful* if its returning set has the properties described in Lemma 1. The omitted proofs in this section can be found in Appendix **??**.

---

**Algorithm 1** BASICMW

1: Let the initial weight $w_i^{(0)} \leftarrow 1$ for each player $i \in \{1, 2, \ldots, k\}$.
2: Let $T \leftarrow 10 \ln k$.
3: **for** $t \leftarrow 0$ **to** $T - 1$ **do**
4:    Let $p^{(t)}(i) \leftarrow \frac{w_i^{(t)}}{\sum_{i=1}^{k} w_i^{(t)}}$ for each $i \in \{1, 2, \ldots, k\}$ so that $p^{(t)}(\cdot)$ defines a probability distribution.
5:    Let $D^{(t)} \leftarrow \sum_{i=1}^{K} p^{(t)}(i) D_i$.
6:    Let $S^{(t)}$ be a set of $\mathbb{S}_{\frac{\epsilon}{120}, \frac{\delta}{4(t+1)^2}}$ samples from $D^{(t)}$. Let $g^{(t)} \leftarrow \mathcal{L}_{\frac{\epsilon}{120}, \frac{\delta}{4(t+1)^2}, \mathcal{F}}(S^{(t)})$.
7:    Let $Z^{(t)} \leftarrow \text{TEST}(g^{(t)}, k, t, \epsilon, \delta)$.
8:    **for each** $i \in \{1, 2, \ldots, k\}$ **do**
9:       **if** $i \in Z^{(t)}$ **then**
10:          $w_i^{(t+1)} \leftarrow w_i^{(t)}$
11:       **else**
12:          $w_i^{(t+1)} \leftarrow 2 \cdot w_i^{(t)}$.
13: **return** $g = \text{Plurality}(g^{(0)}, \ldots, g^{(T-1)})$.

---

**Algorithm 2** Accuracy Test ($\text{TEST}(g, k, t, \epsilon, \delta)$)

1: **for each** $i \in \{1, 2, \ldots, k\}$ **do** Let $T_i$ be a set of $\frac{432}{\epsilon} \ln \left( \frac{k \cdot 4(t+1)^2}{\delta} \right)$ samples from $D_i$.
2: **return** $\{i \mid \text{err}_{T_i}(g) \leq \frac{\epsilon}{6}\}$.

**Lemma 1** *With probability at least* $1 - \frac{\delta}{4(t+1)^2}$, *TEST*$(g, k, t, \epsilon, \delta)$ *returns a set of players that includes 1) each $i$ such that* $\mathrm{err}_{D_i}(g) \leq \frac{\epsilon}{12}$, 2) *none of the $i$ such that* $\mathrm{err}_{D_i}(g) > \frac{\epsilon}{4}$.

Given a function $g$ and a distribution $D$, we say that $g$ is a *good candidate* for $D$ if $\mathrm{err}_D(g) \leq \frac{\epsilon}{4}$. The following lemma shows that if we have a set of functions where most of them are good candidates for $D$, then the plurality vote of these functions also has good accuracy for $D$.

**Lemma 2** *Let $g_1, g_2, \ldots, g_m$ be a set of functions such that more than $70\%$ of them are good candidates for $D$. Let $g = \mathrm{Plurality}(g_1, g_2, \ldots, g_m)$, we have that* $\mathrm{err}_D(g) \leq \epsilon$.

We let the $\mathcal{E}$ be the event that every call of the learner $\mathcal{L}$ and TEST is successful. It is straightforward to see that

$$\mathbf{Pr}[\mathcal{E}] \geq 1 - \sum_{t=0}^{+\infty} \frac{\delta}{4(t+1)^2} \cdot 2 = 1 - \frac{\delta \cdot \pi^2}{24} > 1 - \delta. \tag{1}$$

Now we are ready to prove the main theorem for Algorithm 1.

**Theorem 3** *Algorithm 1 has the following properties.*

1. *With probability at least $1 - \delta$, it returns a function $g$ such that $\mathrm{err}_{D_i}(g) \leq \epsilon$ for all $i \in \{1, 2, \ldots, k\}$.*

2. *Its sample complexity is $O\left( \frac{\ln k}{\epsilon}(d + k \ln \delta^{-1} + k \ln k) \right)$.*

*Proof.* While the sample complexity is easy to verify, we focus on the proof of the first property. In particular, we show that when $\mathcal{E}$ happens (which is with probability at least $1 - \delta$ by (1)), we have $\mathrm{err}_{D_i}(g) \leq \epsilon$ for all $i \in \{1, 2, \ldots, k\}$.

For now till the end of the proof, we assume that $\mathcal{E}$ happens.

For each round $t$, we have that $\frac{\epsilon}{120} \geq \mathrm{err}_{D^{(t)}}(g^{(t)}) = \mathbb{E}_{i \sim p^{(t)}(\cdot)}[\mathrm{err}_{D_i}(g^{(t)})]$. Therefore, by Markov inequality, we have that $\mathbf{Pr}_{i \sim p^{(t)}(\cdot)}\left[\mathrm{err}_{D_i}(g^{(t)}) > \frac{\epsilon}{12}\right] \leq .1$. In other words,

$$.1 \geq \sum_{i:\mathrm{err}_{D_i}(g^{(t)}) > \frac{\epsilon}{12}} p^{(t)}(i) = \frac{1}{\sum_{i=1}^{k} w_i^{(t)}} \sum_{i:\mathrm{err}_{D_i}(g^{(t)}) > \frac{\epsilon}{12}} w_i^{(t)}. \tag{2}$$

Now consider the total weight $\sum_{i=1}^{k} w_i^{(t+1)}$, we have

$$\sum_{i=1}^{k} w_i^{(t+1)} = \sum_{i=1}^{k} w_i^{(t)} + \sum_{i \notin Z^{(t)}} w_i^{(t+1)}. \tag{3}$$

By Lemma 1 and $\mathcal{E}$, we have that

$$\sum_{i \notin Z^{(t)}} w_i^{(t+1)} \leq \sum_{i:\mathrm{err}_{D_i}(g^{(t)}) > \frac{\epsilon}{12}} w_i^{(t+1)}. \tag{4}$$

Combining (2), (3), and (4), we have $\sum_{i=1}^{k} w_i^{(t+1)} \leq 1.1 \sum_{i=1}^{k} w_i^{(t)}$. Since $\sum_{i=1}^{k} w_i^{(0)} = k$, we have the following inequality holds for every $t = 0, 1, 2, \ldots$ : $\sum_{i=1}^{k} w_i^{(t)} \leq 1.1^t \cdot k$.

Now let us focus on an arbitrary player $i$. We will show that for at least $70\%$ of the rounds $t$, we have $\mathrm{err}_{D_i}(g^{(t)}) \leq \frac{\epsilon}{4}$, and this will conclude the proof of this theorem thanks to Lemma 2.

Suppose the contrary: for more than $30\%$ of the rounds, we have $\mathrm{err}_{D_i}(g^{(t)}) > \frac{\epsilon}{4}$. At each of such round $t$, we have $i \notin Z^{(t)}$ because of Lemma 1 and $\mathcal{E}$, and therefore $w_i^{(t+1)} = 2 \cdot w_i^{(t)}$. Therefore, we have $w_i^{(T)} \geq 2^{.3T}$. Together with (4), we have $2^{.3T} \leq w_i^T \leq \sum_{i=1}^{k} w_i^{(T)} \leq 1.1^T \cdot k$, which is a contradiction for $T = 10 \ln k$. $\square$

# 3 The Quest for Optimality via Robust Multiplicative Weights

In this section we improve the result in Theorem 3 to get an optimal algorithm when $k$ is polynomially bounded by $d$ (see Theorem 4; the optimality will be shown in Section 4). In fact, our improved algorithm (Algorithm 3 using Algorithm 4 as a subroutine), is almost the same as Algorithm 1 (using Algorithm 2 as a subroutine). We highlight the differences as follows.

1. The total number of iterations at Line 2 of Algorithm 1 is changed to $\tilde{T} = 2000 \ln(k/\delta)$.

2. The failure probability for the single-distribution learning algorithm $\mathcal{L}$ at Line 6 of Algorithm 1 is increased to a constant $1/100$.

3. The number of times that each distribution is sampled at Line 1 of Algorithm 2 is reduced to $\frac{432}{\epsilon} \ln(100)$.

Although these changes seem minor, it requires substantial technical efforts to establish Theorem 4. We describe the challenge and sketch our solution as follows.

While the 2nd and 3rd items lead to the key reduction of the sample complexity, they make it impossible to use the union bound and claim that with high probability "every call of $\mathcal{L}$ and TEST is successful" (see Inequality (1) in the analysis for Algorithm 1).

To address this problem, we will make our multiplicative weight analysis robust against occasionally failed rounds so that it works when "most calls of $\mathcal{L}$ and WEAKTEST are successful".

In more details, we will first work on the total weights $W^{(t)} = \sum_{i=1}^{k} w_i^{(t)}$ at the $t$-th round, and show that conditioned on the $t$-th round, $\mathbb{E}[W^{(t+1)}]$ is upper bounded by $1.13W^{(t)}$ (where in contrast we had a stronger and deterministic statement $\sum_{i=1}^{k} w_i^{(t+1)} \leq 1.1 \sum_{i=1}^{k} w_i^{(t)}$ in the analysis for the basic algorithm). Using Jensen's inequality we will be able to derive that $\mathbb{E}[\ln W^{(t+1)}]$ is upper bounded by $(\ln 1.13 + \ln W^{(t)})$. Then, using Azuma's inequality for supermartingale random variables, we will show that with high probability, $\ln W^{(\tilde{T})} \leq \tilde{T}(\ln 1.18) + \ln W^{(0)}$, i.e. $W^{(\tilde{T})} \leq 1.18^{\tilde{T}} \cdot k$, which corresponds to $\sum_{i=1}^{k} w_i^{(t)} \leq 1.1^t \cdot k$ in the basic proof. On the other hand, recall that in the basic proof we had to show that if for more than 30% of the rounds, the $g^{(t)}$ function is not a good candidate for a player distribution $D_i$, then we have $w_i^{(T)} \geq 2^{.3T}$. In the analysis for the improved algorithm, because the WEAKTEST procedure fails with much higher probability, we need to use concentration inequalities and derive a slightly weaker statement ($w_i^{(\tilde{T})} \geq 2^{.25\tilde{T}}$). Finally, we will put everything together using the same proof via contradiction argument, and prove the following theorem.

---

**Algorithm 3** MWEIGHTS

1: Let the initial weight $w_i^{(0)} \leftarrow 1$ for each player $i \in \{1, 2, 3, \ldots, k\}$.
2: Let $\tilde{T} \leftarrow 2000 \ln(k/\delta)$.
3: **for** $t \leftarrow 0$ **to** $\tilde{T} - 1$ **do**
4:     Let $p^{(t)}(i) \leftarrow \frac{w_i^{(t)}}{\sum_{i=1}^{k} w_i^{(t)}}$ for each $i \in \{1, 2, 3, \ldots, k\}$ so that $p^{(t)}(\cdot)$ defines a probability distribution.
5:     Let $D^{(t)} \leftarrow \sum_{i=1}^{K} p^{(t)}(i)D_i$.
6:     Let $S^{(t)}$ be a set of $\mathbb{S}_{\frac{\epsilon}{120}, \frac{1}{100}}$ samples from $D^{(t)}$. Let $g^{(t)} \leftarrow \mathcal{L}_{\frac{\epsilon}{120}, \frac{1}{100}, \mathcal{F}}(S^{(t)})$.
7:     Let $Z^{(t)} \leftarrow$ WEAKTEST$(k, g^{(t)}, \epsilon, \delta)$.
8:     **for each** $i \in \{1, 2, 3, \ldots, k\}$ **do**
9:         **if** $i \in Z^{(t)}$ **then**
10:             $w_i^{(t+1)} \leftarrow w_i^{(t)}$
11:         **else**
12:             $w_i^{(t+1)} \leftarrow 2 \cdot w_i^{(t)}$.
13: **return** $g = \text{Plurality}(g^{(0)}, \ldots, g^{(\tilde{T}-1)})$.

---

**Algorithm 4** Weak Accuracy Test (WEAKTEST$(g, k, \epsilon, \delta)$)

1: **for each** $i \in \{1, 2, 3, \ldots, k\}$ **do** Let $T_i$ be a set of $\frac{432}{\epsilon} \ln(100)$ samples from $D_i$.
2: **return** $\{i \mid \text{err}_{T_i}(g) \leq \frac{\epsilon}{6}\}$.

---

**Theorem 4** *Algorithm 3 has the following properties.*

1. *With probability at least $1 - \delta$, it returns a function $g$ such that $\text{err}_{D_i}(g) \leq \epsilon$ for all $i \in \{1, 2, \ldots, k\}$.*

2. *Its sample complexity is $O\left( \frac{(\ln k + \ln \delta^{-1})(d + k)}{\epsilon} \right)$.*

Now we prove Theorem 4.

Similarly to Lemma 1, applying Proposition **??** (but without the union bound), we have the following lemma for WEAKTEST.

**Lemma 5** *For each player $i$, with probability at least $1 - \frac{1}{100}$, the following hold, 1) if $\mathrm{err}_{D_i}(g) \leq \frac{\epsilon}{12}$, then $i \in$ WEAKTEST$(g, k, \epsilon, \delta)$; 2) if $\mathrm{err}_{D_i}(g) > \frac{\epsilon}{4}$, then $i \notin$ WEAKTEST$(g, k, \epsilon, \delta)$.*

Let the indicator variable $\psi_i^{(t)} = 1$ if the desired event described in Lemma 5 for $i$ and time $t$ does not happen; and let $\psi_i^{(t)} = 0$ otherwise. By Lemma 5, we have $\mathbb{E}[\psi_i^{(t)}] \leq \frac{1}{100}$. By Proposition **??**, for each player $i$, we have $\mathbf{Pr}\left[\sum_{t=0}^{\tilde{T}-1} \psi_i^{(t)} > .05\tilde{T}\right] \leq \exp\left(-\frac{1}{3} \cdot 4^2 \cdot \frac{\tilde{T}}{100}\right) \leq \exp\left(-\frac{5\tilde{T}}{100}\right) \leq \frac{\delta}{k^5}$.

Now let $\mathcal{J}_1$ be the event that $\sum_{t=0}^{\tilde{T}-1} \psi_i^{(t)} \leq .05\tilde{T}$ for every $i$. Via a union bound, we have that

$$\mathbf{Pr}[\mathcal{J}_1] \geq 1 - \frac{\delta}{k^4}. \tag{5}$$

Let the indicator variable $\chi^{(t)} = 1$ if the learner $\mathcal{L}$ fails at time $t$; and let $\chi^{(t)} = 0$ otherwise. We have

$$\mathbb{E}\left[\chi^{(t)} \mid \text{time } 0, 1, \ldots, t-1\right] \leq \frac{1}{100}. \tag{6}$$

Let $W^{(t)} = \sum_{i=1}^{k} w_i^{(t)}$ be the total weights at time $t$. For each $t$, similarly to (3), we have

$$W^{(t+1)} = W^{(t)} + \sum_{i \notin Z^{(t)}} w_i^{(t)}. \tag{7}$$

For each $i$ such that $\mathrm{err}_{D_i}(g^{(t)}) \leq \frac{\epsilon}{12}$, by Lemma 5, we know that $\mathbf{Pr}[i \notin Z^{(t)}] \leq \frac{1}{100}$. Therefore, if we take the expectation over the randomness of WEAKTEST at time $t$, we have,

$$\mathbb{E}\left[\sum_{i \notin Z^{(t)}} w_i^{(t)}\right] \leq \sum_{i:\mathrm{err}_{D_i}(g^{(t)})>\frac{\epsilon}{12}} w_i^{(t)} + \mathbb{E}\left[\sum_{i:\mathrm{err}_{D_i}(g^{(t)})\leq\frac{\epsilon}{12}} w_i^{(t)}\right]$$

$$\leq \sum_{i:\mathrm{err}_{D_i}(g^{(t)})>\frac{\epsilon}{12}} w_i^{(t)} + \frac{1}{100} \cdot \sum_{i=1}^{k} w_i^{(t)}. \tag{8}$$

When $\chi^{(t)} = 0$, similarly to the proof of Theorem 3, we have $\mathbf{Pr}_{i \sim p^{(t)}(\cdot)}\left[\mathrm{err}_{D_i}(g^{(t)}) > \frac{\epsilon}{12}\right] \leq .1$, and

$$.1 \geq \sum_{i:\mathrm{err}_{D_i}(g^{(t)})>\frac{\epsilon}{12}} p^{(t)}(i) = \frac{1}{\sum_{i=1}^{k} w_i^{(t)}} \sum_{i:\mathrm{err}_{D_i}(g^{(t)})>\frac{\epsilon}{12}} w_i^{(t)}. \tag{9}$$

Combining (7), (8), and (9), we have (when $\chi^{(t)} = 0$)

$$\mathbb{E}\left[W^{(t+1)} \mid \chi^{(t)} = 0 \text{ and } W^{(0)}, \ldots, W^{(t)}\right] \leq 1.11 \cdot W^{(t)}. \tag{10}$$

Together with (6), we have $\mathbb{E}\left[W^{(t+1)} \mid W^{(0)}, \ldots, W^{(t)}\right] \leq 1.11 \cdot W^{(t)} = \ln\left(\mathbb{E}\left[W^{(t+1)} \mid \chi^{(t)} = 0 \text{ and } W^{(0)}, \ldots, W^{(t)}\right] \cdot \mathbf{Pr}\left[\chi^{(t)} = 0 \mid W^{(0)}, \ldots, W^{(t)}\right] + 2W^{(t)} \cdot \mathbf{Pr}\left[\chi^{(t)} = 1 \mid W^{(0)}, \ldots, W^{(t)}\right]\right) \leq (1.11 + 0.02)W^{(t)} = 1.13W^{(t)}$.

Let $Q^{(t)} = \ln W^{(t+1)}/W^{(t)}$, and by Jensen's inequality, we have $\mathbb{E}\left[Q^{(t)} \mid W^{(0)}, \ldots, W^{(t)}\right] \leq \ln \mathbb{E}\left[W^{(t+1)}/W^{(t)} \mid W^{(0)}, \ldots, W^{(t)}\right]$. Therefore, we have $\mathbb{E}\left[Q^{(t)} \mid Q^{(0)}, \ldots, Q^{(t-1)}\right] = \mathbb{E}\left[Q^{(t)} \mid W^{(0)}, \ldots, W^{(t)}\right] \leq \ln \mathbb{E}\left[W^{(t+1)}/W^{(t)} \mid W^{(0)}, \ldots, W^{(t)}\right] \leq \ln(1.11 + .02) = \ln 1.13$.

Now let $\tilde{Q}^{(t)} = \sum_{z=0}^{t-1} Q^{(z)} - t \cdot \ln 1.13$ for all $t = 0, 1, 2, \ldots$. We have that $\{\tilde{Q}^{(t)}\}$ is a super-martingale and $|\tilde{Q}^{(t+1)} - \tilde{Q}^{(t)}| \leq \ln 2$ for all $t = 0, 1, 2, \ldots$. By Proposition **??** and noticing that

$\ln 1.18 - \ln 1.13 > .04$, we have $\mathbf{Pr}\left[\sum_{t=0}^{\tilde{T}-1} Q^{(t)} > (\ln 1.18)\tilde{T}\right] \leq \mathbf{Pr}\left[\tilde{Q}^{(\tilde{T})} - \tilde{Q}^{(0)} > .04\tilde{T}\right] \leq$
$\exp\left(-\frac{.04^2 \cdot \tilde{T}}{2 \cdot (\ln 2)^2}\right) \leq \frac{\delta}{k^2}$. Let $\mathcal{J}_2$ be the event that $W^{(\tilde{T})} \leq 1.18^{\tilde{T}} \cdot k \Leftrightarrow \sum_{t=0}^{\tilde{T}-1} Q^{(t)} \leq (\ln 1.18)\tilde{T}$,
we have that

$$\mathbf{Pr}[\mathcal{J}_2] \geq 1 - \frac{\delta}{k^2}. \tag{11}$$

Now let $\mathcal{J} = \mathcal{J}_1 \cap \mathcal{J}_2$, combining (5) and (11), for $k \geq 2$, we have

$$\mathbf{Pr}[\mathcal{J}] \geq 1 - \frac{\delta}{k}. \tag{12}$$

Now we are ready to prove Theorem 4 for Algorithm 3.

*Proof.* [of Theorem 4] While the sample complexity is easy to verify, we focus on the proof of the first property. In particular, we show that when $\mathcal{J}$ happens (which is with probability at least $1 - \delta$ by (12)), we have $\mathrm{err}_{D_i}(g) \leq \epsilon$ for all $i \in \{1, 2, 3, \ldots, k\}$.

Let us consider an arbitrary player $i$. We will show that when $\mathcal{J}$ happens, for at least $70\%$ the times $t$, we have $\mathrm{err}_{D_i}(g^{(t)}) \leq \frac{\epsilon}{4}$, and this will conclude the proof of this theorem thanks to Lemma 2.

Suppose the contrary: for more than $30\%$ of the times, we have $\mathrm{err}_{D_i}(g^{(t)}) > \frac{\epsilon}{4}$. Because of $\mathcal{J}_1$, for more than $30\% - 5\% = 25\%$ of the times $t$, we have $i \notin Z^{(t)}$. Therefore, we have $w_i^{(\tilde{T})} \geq 2^{.25\tilde{T}}$. On the other hand, by $\mathcal{J}_2$ we have $W^{(\tilde{T})} \leq 1.2^{\tilde{T}}$. Therefore, we reach $2^{.25\tilde{T}} \leq w_i^{(\tilde{T})} \leq W^{(\tilde{T})} \leq 1.18^{\tilde{T}} \cdot k$, which is a contradiction to $\tilde{T} = 2000 \ln(k/\delta)$. $\qquad\square$

# 4 Lower Bound

We show the following lower bound result, which matches our upper bound (Theorem 3) when $k = (1/\delta)^{\Omega(1)}$ and $k = d^{O(1)}$.

**Theorem 6** *In collaborative PAC learning with $k$ players and a hypothesis class of VC-dimension $d$, for any $\epsilon, \delta \in (0, 0.01)$, there exists a hard input distribution on which any $(\epsilon, \delta)$-learning algorithm $\mathcal{A}$ needs $\Omega(\max\{d \ln k, k \ln d\}/\epsilon)$ samples in expectation, where the expectation is taken over the randomness used in obtaining the samples and the randomness used in drawing the input from the input distribution.*

The proof of Theorem 6 is similar to that for the lower bound result in [3]; however, we need to generalize the hard instance provided in [3] in two different cases. We briefly discuss the high level ideas of our generalization here, and leave the full proof to Appendix **??** due to space constraints.

The lower bound proof in [3] (for $k = d$) performs a reduction from a simple player problem to a $k$-player problem, such that if we can $(\epsilon, \delta)$-PAC learn the $k$-party problem using $m$ samples in total, then we can $(\epsilon, 10\delta/(9k))$-PAC learn the single player problem using $O(m/k)$ samples. Now for the case when $d > k$, we need to change the single player problem used in [3] whose hypothesis class is of VC-dimension $\Theta(1)$ to one whose hypothesis class is of VC-dimension $\Theta(d/k)$. For the case when $d \leq k$, we essentially duplicate the hard instance for a $d$-player problem $k/d$ times, getting a hard instance for a $k$-player problem, and then perform the random embedding reduction from the single player problem to the $k$-player problem. See Appendix **??** for details.

# 5 Experiments

We present in this section a set of experimental results which demonstrate the effectiveness of our proposed algorithms.

Our algorithms are based on the assumption that given a hypothesis class, we are able to compute its VC dimension $d$ and access an oracle to compute an $(\epsilon, \delta)$-classifier with sample complexity $\mathbb{S}_{\epsilon, \delta}$. In practice, however, it is usually computationally difficult to compute the exact VC dimension for a

given hypothesis class. Also, the VC dimension usually only proves to be a very loose upper bound for the sample complexity needed for an $(\epsilon, \delta)$-classifier.

To address these practical difficulties, in our experiment, we treat the VC dimension $d$ as a parameter to control the sample budget. More specifically, we will first choose a concrete model as the oracle; in our implementation, we choose the decision tree. We then set the parameter $\delta = 0.9$ and gradually increase $d$ to determine the sample budget. For each fixed sample budget (i.e., each fixed $d$), we run the algorithm for 100 times and test whether the following happens,

$$\widehat{\mathbf{Pr}}[\max_i \text{err}_{D_i}(g) \leq \epsilon \text{ for all } i] \geq 0.9. \tag{13}$$

Here $\epsilon$ is a parameter we choose and $g$ is the classifier returned by the collaborative learning algorithm to be tested. The empirical probability $\widehat{\mathbf{Pr}}[\cdot]$ in (13) is calculated over the 100 runs. We finally report the minimum number of samples consumed by the algorithm to achieve (13).

Note that in our theoretical analysis, we did not try to optimize the constants. Instead, we tune the constants for both CENLEARN and MWEIGHTS for better performance. Please find more implementation details in the appendix.

**Datasets.**  We will test the collaborative learning algorithms using the following data sets.

MAGIC-EVEN [4]. This data set is generated to simulate registration of high energy gamma particles in an atmospheric Cherenkov telescope. There are $19,020$ instances and each belongs to one of the two classes (gamma and hadron). There are 11 attributes in each data point. We randomly partition this data set into $k = 10$ subsets (namely, $D_1, \ldots, D_k$).

MAGIC-1. The raw data set is the same as we have in MAGIC-EVEN. Instead of random partitioning, we partition the data set into $D_1$ and $D_2$ based on the two different classes, and make $k - 2$ more copies of $D_2$ so that $D_2, D_3, \ldots, D_k$ are identical. In our case we set $k = 10$.

MAGIC-2. This data set differs from MAGIC-1 in the way of constructing $D_1$ and $D_2$: we partition the original data set into $D_1$ and $D_2$ based on the first dimension of the feature vectors; we then make duplicates for $D_2$. Here we again set $k = 10$.

WINE [5]. This data set contains physicochemical tests for white wine, and the scores of the wine range from 0 to 10. There are $4,898$ instances and there are 12 attributes in the feature vectors. We partition the data set into $D_1, \ldots, D_4$ based on the first two dimensions.

EYE. This data set consists of 14 EEG values and a value indicating the eye state. There are $14,980$ instances in this data set. We partition it into $D_1, \ldots, D_4$ based on the first two dimensions.

LETTER [8]. This data set has $20,000$ instances, each in $\mathbb{R}^{16}$. There are 26 classes, each representing one of 26 capital letters. We partition this data set into $k = 12$ subsets based on the first 4 dimensions of the feature vectors.

**Tested Algorithms.**  We compare our algorithms with the following two baseline algorithms,

NAIVE. In this algorithm we treat all distributions $D_1, \ldots, D_k$ equally. That is, given a budget $z$, we sample $z$ training samples from $D = \frac{1}{k} \sum_{i=1}^{k} D_i$. We then train a classifier (decision tree) using those samples.

CENLEARN, this is the implementation of the algorithm proposed by Blum et al. [3].

Since our Algorithm 1 and Algorithm 3 are very similar, and Algorithm 3 has better theoretical guarantee, we will only test Algorithm 3, denoted as MWEIGHTS, in our experiments.

**Experimental Results and Discussion.**  The experimental results are presented in Figure 1. We test the algorithms for each data set using multiple values of the error threshold $\epsilon$, and report the sample complexity for NAIVE, MWEIGHTS and CENLEARN.

In Figure 1a, we notice that NAIVE uses less samples than its competitors. This phenomenon is predictable because in MAGIC-EVEN, $D_1, \ldots, D_k$ are constructed via random partitioning, which is the easiest case for NAIVE. Since MWEIGHTS and CENLEARN need to train multiple classifiers, each classifier will get fewer training samples than NAIVE when the total budgets are the same.

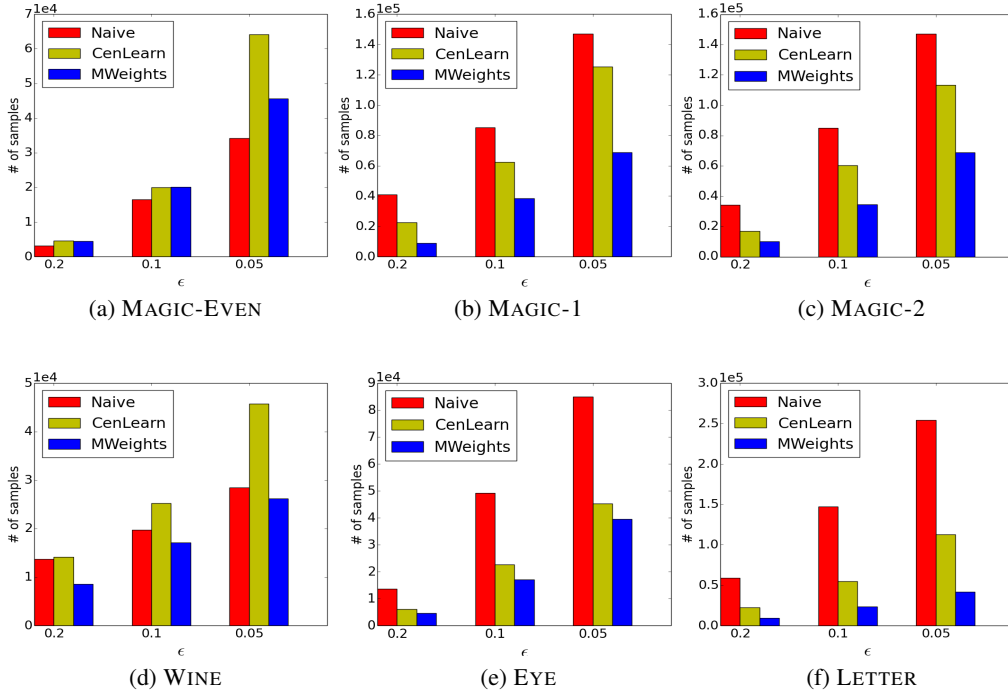

Figure 1: Sample complexity versus error threshold $\epsilon$.

In Figure 1b and Figure 1c, $D_1, \ldots, D_k$ are constructed in a way that $D_2, D_3, \ldots, D_k$ are identical, and $D_1$ is very different from other distributions. Thus the overall distribution (i.e., $D = \frac{1}{k} \sum_{i=1}^{k} D_i$) used to train NAIVE is quite different from the original data set. One can observe from those two figures that MWEIGHTS still works quite well while NAIVE suffers.

In Figure 1b-Figure 1f, one can observe that MWEIGHTS uses fewer samples than its competitors in almost all cases, which shows the superiority of our proposed algorithm. CENLEARN outperforms NAIVE in general. However, NAIVE uses slightly fewer samples than CENLEARN in some cases (e.g., Figure 1d). This may due to the fact that the distributions $D_1, \ldots, D_k$ in those cases are not hard enough to show the superiority of CENLEARN over NAIVE.

To summarize, our experimental results show that MWEIGHTS and CENLEARN need fewer samples than NAIVE when the input distributions $D_1, \ldots, D_k$ are sufficiently different. MWEIGHTS consistently outperforms CENLEARN, which may due to the facts that MWEIGHTS has better theoretical guarantees and is more straightforward to implement.

## 6  Conclusion

In this paper we consider the collaborative PAC learning problem. We have proved the optimal overhead ratio and sample complexity, and conducted experimental studies to show the superior performance of our proposed algorithms.

One open question is to consider the *balance* of the numbers of queries made to each player, which can be measured by the ratio between the largest number of queries made to a player and the average number of queries made to the $k$ players. The proposed algorithms in this paper may attain a balance ratio of $\Omega(k)$ in the worst case. It will be interesting to investigate:

1. Whether there is an algorithm with the same sample complexity but better balance ratio?
2. What is the optimal trade-off between sample complexity and balance ratio?

**Acknowledgments**

Jiecao Chen and Qin Zhang are supported in part by NSF CCF-1525024, CCF-1844234 and IIS-1633215. Part of the work was done when Yuan Zhou was visiting the Shanghai University of Finance and Economics.

## Footnotes

[2]We note that this is a stronger statement than the earlier one on the "the optimal overhead ratio of $O(\ln k)$ for $k = O(d)$" in several aspects. First, the showing the optimal overhead ratio only needs a minimax lower bound; while in the latter statement we claim the optimal sample complexity for every $k$ and $d$ in the range. Second, our latter statement works for a much wider parameter range for $k$ and $d$.

[3]I.e. the most frequent value, where ties broken arbitrarily.

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
