[Supplementary Material]

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

# A Concentration Bounds

**Proposition 7 (Multiplicative Chernoff bound)** *Let $X_i (1 \leq i \leq n)$ be independent random variables with values in $[0,1]$. Let $X = \frac{1}{n} \sum_{i=1}^{n} X_i$. For every $0 \leq \epsilon \leq 1$, we have that*

$$\mathbf{Pr}\big[X < (1-\epsilon)\,\mathbb{E}[X]\big] < \exp\left(-\frac{\epsilon^2 n\,\mathbb{E}[X]}{2}\right),$$

$$\mathbf{Pr}\big[X > (1+\epsilon)\,\mathbb{E}[X]\big] < \exp\left(-\frac{\epsilon^2 n\,\mathbb{E}[X]}{3}\right).$$

**Definition 8 (Supermartingale Random Variables)** *A discrete-time* supermartingale *is a sequence of random variables $X_0, X_1, X_2, \dots$ that satisfies for any time $t$,*

$$\mathbb{E}\,|X_t| < \infty, \text{ and } \mathbb{E}[X_{t+1}|X_0, \dots, X_t] \leq X_t.$$

**Proposition 9 (Azuma's inequality for supermartingale random variables)** *Suppose $\{X_k : k = 0, 1, 2, \dots\}$ is a supermartingale and $|X_k - X_{k+1}| \leq c_k$ almost surely. Then for all positive integers $T$ and all positive reals $\theta$,*

$$\mathbf{Pr}[X_T - X_0 \geq \theta] \leq \exp\left(\frac{-\theta^2}{2\sum_{k=0}^{T-1} c_k^2}\right).$$

# B Omitted Proofs in Section 2

*Proof.* [of Lemma 1] For each $i$ such that $\mathrm{err}_{D_i}(g) \leq \frac{\epsilon}{12}$, by Proposition 7, we have that

$$\mathbf{Pr}\left[\mathrm{err}_{T_i}(g) > \frac{\epsilon}{2}\right] \leq \exp\left(-\frac{432}{\epsilon}\ln\left(\frac{k \cdot 4(t+1)^2}{\delta}\right) \cdot \frac{\epsilon}{12^2 \cdot 3}\right) = \frac{\delta}{k \cdot 4(t+1)^2}.$$

Therefore, with probability at least $1 - \frac{\delta}{k \cdot 4(t+1)^2}$, $i$ is included in the output of TEST.

Similarly, for each $i$ such that $\mathrm{err}_{D_i}(g) > \frac{\epsilon}{4}$, by Proposition 7, we have that

$$\mathbf{Pr}\left[\mathrm{err}_{T_i}(g) \leq \frac{\epsilon}{2}\right] \leq \exp\left(-\frac{432}{\epsilon}\ln\left(\frac{k \cdot 4(t+1)^2}{\delta}\right) \cdot \frac{\epsilon}{12^2 \cdot 2}\right) \leq \frac{\delta}{k \cdot 4(t+1)^2}.$$

Therefore, with probability at least $1 - \frac{\delta}{k \cdot 4(t+1)^2}$, $i$ is not included in the output of TEST.

The lemma is now proved by a union bound over at most $k$ players. $\qquad\square$

*Proof.* [of Lemma 2] Suppose for contradiction that $\mathrm{err}_D(g) > \epsilon$. Given a sample $(x,y) \sim D$, when $g(x) \neq y$, we know that for more than half of the $g_i$'s, we have $g_i(x) \neq y$. Therefore, we have

$$\sum_{i=1}^{m} \mathbf{Pr}_{(x,y)\sim D}[g(x) \neq y \text{ and } g_i(x) \neq y] > \frac{\epsilon m}{2}. \tag{14}$$

On the other hand, by discussing whether $g_i$ is a good candidate for $D$, we have

$$\sum_{i=1}^{m} \mathbf{Pr}_{(x,y)\sim D}[g(x) \neq y \text{ and } g_i(x) \neq y]$$
$$\leq \sum_{i:g_i \text{ good}} \mathbf{Pr}_{(x,y)\sim D}[g_i(x) \neq y] + \sum_{i:g_i \text{ not good}} \mathbf{Pr}_{(x,y)\sim D}[g(x) \neq y]$$
$$\leq \sum_{i:g_i \text{ good}} \frac{\epsilon}{4} + \sum_{i:g_i \text{ not good}} \epsilon \leq .7m \cdot \frac{\epsilon}{4} + .3m \cdot \epsilon < .5m\epsilon,$$

which contradicts (14). $\qquad\square$

## C   Proof of Theorem 6

Before proving Theorem 6 we need a result from [6]. Let $\phi_d$ be the following input distribution.

- Instance space $\mathcal{Y}_d = \{0, 1, \ldots, d-1, \perp\}$.
- Hypothesis class: $\mathcal{G}_d$ is the collection of all binary functions on $\mathcal{Y}_d$ that map $\perp$ to 0.
- Target function: $g^*$ is chosen uniformly at random from $\mathcal{G}_d$.
- Player's distribution: $\mathbf{Pr}[\perp] = 1 - 8\epsilon$, and $\mathbf{Pr}[0] = \ldots \mathbf{Pr}[d-1] = 8\epsilon/k$.

**Lemma 10 ([6])** *For any $\epsilon, \delta \in (0, 0.01)$, any $(\epsilon, \delta)$-learning algorithm $\mathcal{A}$ on $\phi_d$ needs $\Omega(d/\epsilon)$ samples in expectation, where the expectation is taken over the randomness used in obtaining the samples and the randomness used in drawing the input from $\phi_d$.*

We prove Theorem 6 in two cases: $d > k$ and $d \leq k$.

**The case $d > k$.**   Let $\sigma(i, j) = (i - 1) \cdot d/k + j$. We create the following hard input distribution, denoted by $\Phi_{k,d}$.

- Instance space: $\mathcal{X}_d = \{0, 1, \ldots, d-1, \perp\}$.
- Hypothesis class: $\mathcal{F}_d$ is the collection of all binary functions on $\mathcal{X}_d$ that map $\perp$ to 0.
- Target function: $f^*$ is chosen uniformly at random from $\mathcal{F}_d$.
- Player $i$'s distribution $D_i$ (for each $i \in [k]$): Assigns weights to items in $\{\sigma(i, 0), \sigma(i, 1), \ldots, \sigma(i, d-1), \perp\}$ as follows: $\mathbf{Pr}[\perp] = 1 - 8\epsilon$, and $\mathbf{Pr}[\sigma(i, 0)] = \ldots \mathbf{Pr}[\sigma(i, d/k - 1)] = 8\epsilon/k$. For any other item $x \in \mathcal{X}_d$, $\mathbf{Pr}[x] = 0$.

Note that the induced input distribution for the $i$-th player is the same as $\phi_{d/k}$ for any $i \in [k]$.

We have the following lemma. It is easy to see that Lemma 11 and Lemma 10 imply a sample complexity $\Omega(d \ln k/\epsilon)$ for any $(\epsilon, \delta)$-learning algorithm on input distribution $\Phi_{k,d}$ in expectation.

**Lemma 11** *If there exists an $(\epsilon, \delta)$-learning algorithm $\mathcal{A}'$ that uses $m$ samples in expectation on input distribution $\Phi_{k,d}$, then there exists an $(\epsilon, \frac{10}{9k} \cdot \delta)$-learning algorithm $\mathcal{A}$ that uses $\frac{10}{9k} \cdot m$ samples in expectation on input distribution $\phi_{d/k}$.*

*Proof.*   We construct $\mathcal{A}'$ for input distribution $\phi_{d/k}$ using $\mathcal{A}$ for input distribution $\Phi_{k,d}$ as follows.

1. $\mathcal{A}'$ draws an input instance $(\mathcal{F}_d, f^*, \{D_i\}_{i \in [k]})$ from $\Phi_{k,d}$, and samples $\ell$ uniformly at random from $[k]$.

2. $\mathcal{A}'$ simulates $\mathcal{A}$ on instance $(\mathcal{F}_d, f^*, \{D_i\}_{i \in [k]})$ with the input distribution of the $\ell$-th player replaced by $\phi_{d/k}$. Every time $\mathcal{A}$ draws a sample from player $i \neq \ell$, $\mathcal{A}'$ does the same (which is free since $\mathcal{A}'$ already knows $(\mathcal{F}_d, f^*, \{D_i\}_{i \in [k]})$), and passes the sample (and its label) to $\mathcal{A}$. Every time $\mathcal{A}$ draws a sample from player $\ell$, $\mathcal{A}'$ samples from distribution $\phi_{d/k}$ instead. Let $(u, v)$ $(u \in \{0, 1, \ldots, d/k - 1, \perp\}, v \in \{0, 1\})$ be the sample. If $u = \perp$ then $\mathcal{A}'$ passes $(\perp, 0)$ to $\mathcal{A}$, otherwise $\mathcal{A}'$ passes $(\sigma(\ell, u), v)$ to $\mathcal{A}$.

3. When $\mathcal{A}$ terminates and returns a function $f$ on $\mathcal{X}_d$, $\mathcal{A}'$ checks whether the error of $f$ on each $D_i$ $(i \neq \ell)$ is no more than $\epsilon$. If yes, $\mathcal{A}'$ returns $f'$ defined as $f'(\perp) = f(\perp)$, and $f'(u) = f(\sigma(\ell, u))$. Otherwise $\mathcal{A}'$ repeats the simulation on a new input instance from $\Phi_{k,d}$.

We have the following claims, whose proofs can be found in [3] for a similar reduction. The two claims finish the proof of Lemma 11.

**Claim 12** $\mathcal{A}'$ *is an* $(\epsilon, \frac{10}{9k} \cdot \delta)$*-learning algorithm on* $\phi_{d/k}$*, where $\delta$ is failure probability of $\mathcal{A}$.*

**Claim 13** $\mathcal{A}'$ *uses at most* $\frac{10}{9k} \cdot m$ *samples in expectation, where $m$ is the sample complexity of $\mathcal{A}$.*

**The case** $d \leq k$. We againt start by constructing a hard input distribution for the $k$ players, denoted by $\Psi_{k,d}$. We first construct a hard input distribution for the first $d$ players. The construction is the same as the one used in [3] for the case $k = d$.

- Instance space: $\mathcal{X}_d = \{1, 2, \ldots, d, \perp\}$.
- Hypothesis class: $\mathcal{F}_d$ is the collection of all binary functions on $\mathcal{X}_d$ that map $\perp$ to 0.
- Target function: $f^*$ is chosen uniformly at random from $\mathcal{F}_d$.
- Player's distribution $D_i$ ($i \in [d]$): with probability $1/2$, the $i$-th player assigns weights to items in $\{1, 2, \ldots, \perp\}$ as $\mathbf{Pr}[\perp] = 1$ and $\mathbf{Pr}[x] = 0$ for all other items $x \in \mathcal{X}_d$; with probability $1/2$, it assigns weights as $\mathbf{Pr}[\perp] = 1 - 2\epsilon$, $\mathbf{Pr}[i] = 2\epsilon$, and $\mathbf{Pr}[x] = 0$ for all other items $x \in \mathcal{X}_d$.

We then assign the same input distribution for the next $d$ players, the next next $d$ players, and so on. In other words, we duplicate the input distribution of the first $d$ players for $k/d$ times. Finally we randomly permute the $k$ players.

Let $\psi$ denote the input distribution of $\Psi_{1,1}$. We have the following lemma.

**Lemma 14 ([3])** *For any $\epsilon, \delta \in (0, 0.01)$, any $(\epsilon, \delta)$-learning algorithm $\mathcal{A}$ on $\psi$ needs $\Omega(\log(1/\delta)/\epsilon)$ samples in expectation, where the expectation is taken over the randomness used in obtaining the samples and the randomness used in drawing the input from $\psi$.*

We use the following reduction.

1. $\mathcal{A}'$ draws an input instance $(\mathcal{F}_d, f^*, \{D_i\}_{i\in[k]})$ from $\Psi_{k,d}$, and samples $\ell$ uniformly at random from $[k]$.

2. $\mathcal{A}'$ simulates $\mathcal{A}$ on instance $(\mathcal{F}_d, f^*, \{D_i\}_{i\in[k]})$ with the input distribution of the $\ell$-th player replaced by $\psi$. Every time $\mathcal{A}$ draws a sample from player $i \neq \ell$, $\mathcal{A}'$ does the same (which is free since $\mathcal{A}'$ already knows $(\mathcal{F}_d, f^*, \{D_i\}_{i\in[k]})$), and passes the sample (and its label) to $\mathcal{A}$. Every time $\mathcal{A}$ draws a sample from player $\ell$, $\mathcal{A}'$ samples from distribution $\psi$ instead. Let $(u, v)$ ($u \in \{1, \perp\}, v \in \{0, 1\}$) be the sample. If $u = \perp$ then $\mathcal{A}'$ passes $(\perp, 0)$ to $\mathcal{A}$, otherwise $\mathcal{A}'$ passes $(\ell, v)$ to $\mathcal{A}$.

3. When $\mathcal{A}$ terminates and returns a function $f$ on $\mathcal{X}_d$, $\mathcal{A}'$ checks whether the error of $f$ on each $D_i$ ($i \neq \ell$) is no more than $\epsilon$. If yes, $\mathcal{A}'$ returns $f'$ defined as $f'(\perp) = f(\perp)$, and $f'(1) = f(\ell)$. Otherwise $\mathcal{A}'$ repeats the simulation on a new input instance from $\Psi_{k,d}$.

Claim 13 still holds for the above reduction. While Claim 12 changes slightly to the following (by replacing $k$ in Claim 12 to $d$).

**Claim 15** $\mathcal{A}'$ *is an* $(\epsilon, \frac{10}{9d} \cdot \delta)$*-learning algorithm for the primitive problem, where $\delta$ is the failure probability of $\mathcal{A}$.*

The proof is very similar to that for Claim 12. The only difference is the following: Let $p_i$ be the probability that on a random input instance sampled from $\Psi_{k,d}$, the function $f$ returned by $\mathcal{A}$ satisfies $\mathrm{err}_{D_\ell}(f) > \epsilon$ and $\mathrm{err}_{D_i}(f) \leq \epsilon$ for any $i \neq \ell$. We now have $\sum_{i\in[k]} p_i \leq k/d \cdot \delta$ (due to the $k/d$ times of duplication of the input distribution for the first $d$ players), instead of $\sum_{i\in[k]} p_i \leq \delta$ as the case for Claim 12. This difference makes the final failure bound to be $\frac{10}{9d} \cdot \delta$ instead of $\frac{10}{9k} \cdot \delta$ compared with Claim 12.

The $\Omega(k \ln d \log(1/\delta)/\epsilon)$ lower bound follows from Lemma 14, Lemma 13 and Lemma 15. $\square$

## D  Experiment Implementation Details

As mentioned, we did not try to optimize constants in our theoretical analysis. In our experiment, we tuned several parameters for both CENLEARN and MWEIGHTS for better empirical performance. In particular, we made the following changes.

- We set $\mathbb{S}_{\epsilon,\delta} = \frac{d + \log \delta^{-1}}{10\epsilon}$.

- In both MWEIGHTS and CENLEARN, we set the number of iterations ($\tilde{T}$ in MWEIGHTS and $t$ in CENLEARN) to $\lceil 10 \log k \rceil$.

- In WEAKTEST (Algorithm 4) of MWEIGHTS and the TEST process in CENLEARN, we only drew $30/\epsilon$ samples from $D_i$ and returned $\{i \mid \mathrm{err}_{T_i}(g) \leq \frac{\epsilon}{2}\}$.