[Reviews · NeurIPS 2018]

Reviewer 1



The problem considered -- the "collaborative PAC learning" setting, recently introduced by Blum et al. to capture settings where one wants to learn a target function with respect *simultaneously* to k different underlying distributions) is quite natural and interesting. This paper nearly settles (effectively settles, for a large range of parameters) an obvious question: " what is the overhead (as a function of k, the number of underlying distributions) in the sample complexity of collaborative PAC learning, compared to vanilla PAC learning?" An easy upper bound is a factor of k. Blum et al. established an upper bound of O(log^2 k) on the overhead factor (for k = O(d), where d is the VC dimension of the concept class to learn), and a lower bound of Omega(log k) (for the specific case k=d). The main contribution of this paper is to provide an O(log k) upper bound (for k=O(d), again; the general upper bound is slightly more complicated for general k) on that ratio; they also generalize the lower bound to hold for any k=d^{O(1)}. The lower bound is aobtained by generalized and "bootstarting" the lower bound construction of Blum et al., with some changes to handle the base case. The main contribution is, in my opinion, the upper bound: where the algorithm of Blum et al. worked in stages, refining the learning by choosing one of the k "players" uniformly at random, this paper uses a natural and more refined approached by cchoosing at each stage one of the k players according to a weighted distribution, where the weights are obtained by a multiplicative-weight-update scheme. My only complaint about the paper is the presentation: namely, the authors spend 2.5 pages on a suboptimal (and very slightly simpler) version of their algorithm, and then run out of space to prove the actual, optimal one. Given the fact that the simpler one is not used by the more complex one, and that the analysis does not really bring anything more (no new technique, and the algorithm itself is not simpler to implement), I'd suggest skipping it altogether and focus directly on the optimal one.

Reviewer 2



Update after rebuttal: As mentioned in my review, the authors should clarify that using the multiplicative weight scheme was suggested by Blum et al. and its overhead was discussed (without proof) to be the same as one shown in Theorem 3 of this paper---overhead log^2(k). For a reference, see the last paragraph of section 3 in http://papers.nips.cc/paper/6833-collaborative-pac-learning.pdf. The technical insight of this paper is, in addition to using multiplicative weight update, to change the "Accuracy Test" to the "Weak Test" to further reduce the samples complexity down to the overhead of log(k). The authors in their rebuttal have ignored this fact. So it's best if the camera-ready version reflects this fact accurately. Before rebuttal: This paper continues the study of collaboration in the context of machine learning and obtains improved same complexity bounds. Background: The collaborative PAC model, introduced by Blum et al (NIPS’17), considers a setting where k players with different distributions D_i's that are all consistent with some unknown function f^* want to (\epsilon, \delta)-learn a classifier for their own distribution. The question Blum et al. asks is what is the total “overhead” over the sample complexity of accomplishing one task, if the players can collaborate. As an example when players do not collaborate, the k tasks have to be performed individually leading to an overhead of O(k). Blum et al. shows that in the personalized setting, where different players can use different classifiers, the overhead is O(log(k)) with k = O(d) . They also considered a centralized setting where all players should use the same functions. In this case, they showed that overhead is O(log(k)^2) when k = O(d). They also showed a matching lower bound of Omega(log(k)) overhead for the case of k=d. Summary: This paper obtains an improved overhead for the centralized setting. Their main result, appearing in section 3, is that one can get an overhead of O(log(k)) in the centralized setting. More precisely, the sample complexity they obtain is: O(\epsilon^-1 (d+k)(ln(k) + ln(1/delta)). There second contribution is to extend the lower bound of Blum et al. to the more general case of k = d^O(1). Lastly, they do a set of experiments comparing their algorithm to a naive algorithm and the Blum et al. algorithm on a number of datasets and show that their algorithms have better empirical performance. Overall, I think this is a nice paper. The problem and the setting it studies is very natural. I think the subject matter can be of interest to the broader community and can be related to other areas including, federated and distributed learning, multi-task learning, and transfer learning. I appreciate that the authors have complemented their theoretical results with the experimental findings. There are a few weaker points in the paper that I explain below. 1. The Basic algorithm: The authors spend quite a bit of space on the basic algorithm that has a log^2(k) overhead in the Centralized setting. My first concern about this is that Blum et al. discussed this algorithm briefly in their paper (see last paragraph section 3 of the Blum et al.) and the fact that it has log^2(k) overhead without going into the details. The reason is that the proof of this algorithm is too similar to the original approach of Blum et al. So, I don’t think the authors should focus so much of their efforts and presentation the paper on this basic algorithm. Secondly, their proof approach appears to be incremental over Blum et al approach. In my opinion the authors should simply state these results without proof (defer the proof to the appendix) and also mention that it was discussed by Blum et al. 2. I think the novel and interesting part of this paper is indeed the algorithm in Section 3. Unfortunately, very little information is given about this proof. I would suggest that the authors expand this part and include the proofs. 3. I'm not sure if the sample complexity in Theorem 4 is accurate. Let k =1, then this is showing a sample complexity of d/\epsilon ln(1/\delta) for one task. But, the best upper bound we know in the PAC setting is 1/\epsilon (d ln(1/\epsilon) + \ln(1/\delta)) with an additional ln(1/\epsilon). I ask the authors to explain in the rebuttal what the correct sample complexity is in this case.

Reviewer 3



Overview: This paper studies the problem of (realizable-case) Collaborative PAC learning. In more detail, let H be an hypothesis class, and let h*\in H be the target concept. Assume we are given an access to k unknown distributions D1,…,Dk, and that our goal is to output an hypothesis h such that w.h.p the loss of h w.r.t *each* of the D_i’s is at most epsilon. The main results in this paper are: 1. An upper bound of O((d+k)(log k + log(1/delta)) / epsilon), were d is the VC dimension of H, and epsilon and delta are the accuracy and confidence parameters. 2. A lower bound of Omega(dlog(k) + klog(d)) for fixed epsilon,delta (epsilon=delta=0.005). The main technical tool is the Multiplicative Weights method, which is applied similarly like in Adaboost (the authors should discuss the connection with Adaboost and boosting in more detail). The high-level idea is simple: at each round a mixture of the D_i’s is being weakly learned, and the weights of the mixture are updated in a multiplicative fashion according to the loss of the weak learner. Similar arguments like in the analysis of Adaboost yield an upper bound that is weaker by a factor of log(k) from 1 above (such a bound was obtained by Blum et al. in a previous work). Most of the technical effort is dedicated to shaving this factor. I think this is a nice result, with a non-trivial variant of Adaboost. The presentation and the english can and should be significantly improved. A comment to the authors: How do the results and techniques in your paper compare to those in “Improved Algorithms for Collaborative PAC Learniing” (link: https://arxiv.org/abs/1805.08356), and also in “On Communication Complexity of Classification Problems” (link: https://arxiv.org/abs/1711.05893), where variants of Adaboost are also heavily exploited to ensure that different parties agree on the same hypothesis which is consistent (or has low error) w.r.t a distributed sample.